# Electrical Properties of the Carbon Nanotube-Reinforced Geopolymer Studied by Impedance Spectroscopy

**DOI:** 10.3390/ma15103543

**Published:** 2022-05-15

**Authors:** Marcin Górski, Paweł Czulkin, Natalia Wielgus, Sławomir Boncel, Anna W. Kuziel, Anna Kolanowska, Rafał G. Jędrysiak

**Affiliations:** 1Department of Structural Engineering, Faculty of Civil Engineering, Silesian University of Technology, Akademicka 5, 44-100 Gliwice, Poland; natala.wielgus@polsl.pl; 2Department of Physical Chemistry and Technology of Polymers, Faculty of Chemistry, Silesian University of Technology, Strzody 9, 44-100 Gliwice, Poland; 3NanoCarbon Group, Department of Organic Chemistry, Bioorganic Chemistry and Biotechnology, Faculty of Chemistry, Silesian University of Technology, Krzywoustego 4, 44-100 Gliwice, Poland; slawomir.boncel@polsl.pl (S.B.); anna.kuziel@polsl.pl (A.W.K.); anna.kolanowska@polsl.pl (A.K.); rafal.jedrysik@polsl.pl (R.G.J.)

**Keywords:** impedance spectroscopy, geopolymer, carbon nanotubes, smart material, Structural Health Monitoring

## Abstract

Geopolymers, recognized as an ecological alternative to cement concrete, are gaining more and more interest from researchers and the construction industry. Due to the registrable electrical conductivity, this material also attracts the interest of other fields of science and industry as a potential functional material. The article discusses the used geopolymer material, created on the basis of metakaolin and waste Cathode Ray Tubes (CRT) glass, reinforced with ultra-long in-house carbon nanotubes (CNT), in the context of its use as a smart material for Structural Health Monitoring. Long in-house made carbon nanotubes were added to enhance the electrical conductivity of the geopolymer. The impedance spectroscopy method was applied to investigate the conductive properties of this material. The paper shows the microscopic and mechanical characteristics of the materials and presents the results of promising impedance spectroscopy tests.

## 1. Introduction

The idea of conductive materials’ application to monitor concrete structures, especially in the scope of strain and damage sensing, is not a new issue. In the 1990s, Chen and Chung, while admixing cut carbon fibers to the concrete cover, estimated the degree of cracking and deformation of the concrete structure [1,2,3].

These works were developed by many research groups, e.g., [4,5,6,7], which were looking for more effective ways to use carbon-based conductive materials in Structural Health Monitoring (SHM). With the appearance of carbon nanomaterials, they have led to attempts to use mainly CNT [8,9] and graphene [10,11] to monitor and protect various types of building structures, not just concrete structures [12].

Due to increasing environmental awareness related to significant CO_2_ emissions, the construction industry began to look for a replacement for cement concrete. Geopolymers and alkali-activated materials (AAM), which do not use cement, require less process water, and replace non-renewable sand and gravel resources with industrial wastes, have become one of the main research directions, allowing the immobilization of environmentally hazardous materials [13,14,15].

Geopolymer seems to be a very promising functional material as it exhibits enhanced electrical properties. This material is gaining more and more interest in various industries, and its advanced applications include monitoring of deformation, temperature, and humidity.

First attempts have also been made to strengthen the electrical properties of geopolymer by adding conductive fillers such as carbon fibers [16], graphene [17], CNT [18], carbon black [19], and graphite [20]. Incorporation of carbon nanotubes into the geopolymer mixture has been so far explored mainly in the direction of improving mechanical properties [21]. An influence on the mechanical performance depends on the type and content of added nanotubes. Da Luz et al. [22] report that the addition of 0.1% of pristine carbon nanotubes into the metakaolin-based geopolymer increases both compressive and flexural strength (by ≈13% and ≈29%, respectively), but the addition of 0.2% decreases strength in comparison to the control sample. By contrast, functionalized CNT increased the strength the most efficiently up to 0.2% weight content (compressive strength increased by ≈47% and flexural by ≈66% in comparison to geopolymer without CNT). Rovnanik et al. [23] examined fly ash-based geopolymer with multi-walled carbon nanotubes (MWCNT). According to tests, compressive strength and modulus of elasticity increased along with the increase of nanomaterial content from 0.00% to 0.15% and then decreased in samples containing 0.20% of nanotubes. Samples containing 0.15% of MWCNT achieved 70% higher compressive strength than samples without nanomaterial. An addition of nanotubes decreased the fracture toughness and fracture energy except for samples containing 0.20% of nanomaterial. Scientists found that the addition of MWCNT reduces the formation of microcracks. By contrast, Azeem et al. [24] report the biggest increase in the strength of fly ash and slag-based geopolymer after the addition of 0.20% of carbon nanotubes. The compressive strength increased by about 20% while flexural strength by about 15% in comparison to the control sample. In the reported test, the compressive strength increased monotonically along with the increase of CNT content while flexural strength generally decreased after the addition of CNT excluding samples containing 0.20% of nanotubes. Azeem et al. [24] claim that a concentration of MWCNT higher than 0.20% would weaken the structure of material leading to a decrease of the strength. A similar result is presented by Maho et al. [25]. Both compressive strength and flexural strength of fly ash-based geopolymer increased after the addition of MWCNT and the biggest increase was noted in samples containing 0.20% of MWCNT. Compressive strength increased by approximately 35% and flexural strength by 26% in comparison to control samples without nanomaterial. The further addition of MWCNT (up to 0.60%) caused a drop in strength. Maho et al. indicate that the high content of MWCNT increases the water demand of a mixture leading to problems in mixing and compacting processes. Abbasi et al. [26] report that the addition of 0.5% of MWCNT leads to the increase of compressive strength of metakaolin-based geopolymer by 32% and flexural strength by 28%. The authors explain that an optimal amount of MWCNT leads to the bridging of the micro-cracks which enhances the strength. Using higher amounts of nanomaterial (1.0% in the described test) is less effective due to the agglomeration of nanotubes. Jittabut et al. [27] report adding much greater amounts of carbon nanotubes into the fly ash-based geopolymer—1, 2, 3, 4, and 5% calculated by mass, as MWCNT to fly ash ratio. Authors noted that the addition of 1% of MWCNT considerably improves the compressive strength of geopolymer both after 7 days (improvement by ≈18–45%) and 28 days (improvement by ≈5–30%) of curing, in dependence on the concentration of activator. The addition of greater amounts of MWCNT caused decrease of strength. The apparent porosity of the material increased monotonically along with the increase of MWCNT content.

The impedance spectroscopy from the early years of its development became an attractive method for the study of construction materials by monitoring their conductivity. In most papers, authors applied the method to cement- or concrete-based materials [28,29]. The spectra of most of the presented materials had a rather simple diffusion involving a Randles-type shape which allowed for visual detection of differences in the material composition and changes during its binding [30]. Since impedance spectroscopy is a non-invading and non-destructive technique, one of the most interesting applications would be intrinsic control of the constructions. A possibility of monitoring the concrete corrosion, i.e., destruction, has been recently demonstrated [31]. However, the rather small number of papers on that topic and lack of a universal approach indicates the high complexity of the problem. The paper presents advancements of using impedance spectroscopy as potential tool in Structural Health Monitoring on the example of a novel and ecological structural material based on geopolymer reinforced with ultra-long, in-house CNT.

## 2. Materials and Methods

### 2.1. Synthesis and Characterization of Ultra-Long CNTs (UL-MWCNTs)

UL-MWCNTs CNTs were chosen as reinforcement of geopolymer due to their high potential to increase its electrical conductivity.

UL-MWCNTs were grown via catalytic chemical vapor deposition (CVD) under optimized conditions. A (760 °C) STF1200 Tube Furnace (Across International, Livingstone, NJ, USA) operating with a preheater (250 °C) and an automated syringe pump served as the reactor. Further, 5.5 wt.% of ferrocene (catalyst precursor and a partial carbon source) in toluene (the main carbon source) (injection rate 2.8 mL h^–1^) was applied as the feedstock. Argon was used as the carrier gas (flow rate 1.8 L min^–1^). The total time of synthesis was 24 h. Scanning electron microscopy (SEM) (Phenom Pro Desktop SEM) (Thermo Fischer Scientific, Warsaw, Poland) equipped with an EDS detector (15 kV) was used to acquire the micrographs of the geopolymer, MWCNTs, and their nanocomposites.

### 2.2. Carbon Nanotube-Reinforced Geopolymer

Metakaolin was delivered by the Astra Technologia Betonu (Straszyn, Poland). It was not subjected to any additional treatment before tests. The exact chemical composition of metakaolin is given in Table 1.

Cathode Ray Tube (CRT) glass was delivered in crushed form by Thornmann Recycling^®^ (Toruń, Poland) and came from discarded monitors and television screens. It was not subjected to any additional treatment before the preparation of samples. The chemical composition of CRT glass (Table 1) was determined by X-ray fluorescence (XRF) analysis by EkotechLAB^®^ (Gdańsk, Poland). The curve describing the distribution of particles size is shown in Figure 1.

The liquid solution of sodium hydroxide was obtained by dissolving sodium hydroxide pellets in demineralized water for at least 24 h before preparation of the mixture. The 10 mol/L sodium hydroxide solution was used for mixture preparation.

The commercial sodium silicate was used as the second activator. The producer provided the following characterization of sodium silicate solution: ratio of SiO_2_ to Na_2_O between 2.4 and 2.6. The minimum content of oxides (SiO_2_ and Na_2_O) was 39%.

Tests were carried out on samples made of three mixtures. The content of carbon nanotubes (CNT) was the only variable factor. The first, control, group did not contain nanotubes. The second and third groups of samples contained carbon nanotubes in the amount of 0.5 wt.% and 1 wt.% respectively. The exact composition of all mixtures is given in Table 2.

In the first step, both activators (sodium hydroxide and sodium silicate) were mixed for 5 min. CNTs were firstly mixed with metakaolin and CRT glass and then activators were added. The ready mixture was placed in the prismatic forms of dimensions 40 mm × 40 mm × 160 mm, covered tightly, and kept at the ambient temperature ~20°C for 28 days. After 28 days, samples were demolded, weighed, measured, and then subjected to strength tests. Each sample was first subjected to the three-point flexural strength test. The obtained pieces were subjected to a compressive strength test. The whole procedure was performed using the Controls^®^ Model 65-L27C12 Serial no 12020060 (Controls, Milan, Italy) machine according to standard EN 196-1 [32].

### 2.3. Impedance Spectroscopy Methodology

All electrical measurements were performed using BioLogic SP150 (Seyssinet-Pariset, France) potentiostat equipped with a low-current module under room temperature and humidity of about 36%. The samples were studied in a dry state, and no effect of humidity on the conductivity properties was observed. The impedance spectra were recorded in galvanostatic mode with *I*_DC_ = 0, and modulation amplitude equal to 1 nA. The frequency range was from 0.1 Hz to 1 MHz with 20 points per decade on logarithmic scale. The measurement cables were connected via two metal handles attached to the sample at the 12.0 cm distance.

## 3. Results

### 3.1. Carbon Nanotube-Reinforced Geopolymer—Mechanical Properties

Flexural and compressive strength results are presented in Table 3. In the control group, two samples were subjected to a flexural strength test and four to a compressive strength test. Series with CNT consisted of one prism so that consequently one sample was subjected to the flexural strength test and two to the compressive strength test. Tests were performed for preliminary assessment of CNT content on the mechanical behavior of the geopolymer. The number of samples was too small for a thorough analysis and extended studies will be necessary in the future.

The highest flexural and compressive strength was obtained by control samples without CNT. The addition of CNT caused a slight decrease of both flexural and compressive strength but measured mechanical characteristics of geopolymer with CNT were still high and good enough to consider this material for structural applications. The flexural strength increased, while the compressive strength decreased slightly with the increase of CNT content from 0.5% to 1.0%.

Each sample before the strength test was measured with the caliper and weighed. The density of samples (presented in Table 4) was calculated by dividing the mass of the sample by its volume. The difference between the density of samples without CNT and containing CNT was negligible (2%). According to the results, the addition of CNT does not significantly influence the density of geopolymer.

### 3.2. Microstructure

Figure 2 represents SEM images of neat geopolymer surface (a). In turn, Figure 2b shows in-house synthesized MWCNTs. Here, the prolonged time of synthesis enabled to obtain the fibrous-like 0.8 mm-long and ca. 60 nm in outer diameter MWCNTs (aspect ratio ca. 15,000). The in-house MWCNTs were synthesized as vertically aligned, free-standing arrays and in the yield of ca. 30% yield per carbon. Upon introduction of the nanotubes into the geopolymer matrix (c,d), a surface of the so-formed nanocomposites could be characterized by an increased number of cracks and splits. At the same time, an insight into those surfaces clearly revealed a concentration-dependent and statistical distribution of the nanotube agglomerates as black, quasi-oval areas with irregular edges. In a few cases, the presence of the agglomerates was found in the split lines which would suggest a weaker binding and generation of the cracks.

### 3.3. Impedance Spectroscopy

The typical impedance spectra are shown in Figure 3. Figure 3a (Bode plot) shows the values of impedance module $Z=\frac{U_{0}}{I_{0}}\;$ and phase shift φ as functions of frequency. The complex plane plot also called the Nyquist plot (Figure 3b) gives a relation between real $Z_{Re}=Z\cdot cos\left(\varphi \right)$ and imaginary $Z_{Im}=Z\cdot sin\left(\varphi \right)$ impedance components.

The impedance of the material is very high which requires very sensitive equipment capable of registering a low AC signal. In our case, the valuable data were obtained at frequencies lower than 100 Hz (Figure 3a).

The registered conductivity of the modified geopolymer material, though it is very small, is a unique feature. Then, it was important to prove that the conductivity is provided by the material and not by the humidity in its pores or on the surface. For that, we studied two identical samples. The first one was dried in a desiccator for 6 h at 60 °C. The second one was dipped in water for 30 min and then dried with a tissue to remove water from the surface. The resulting impedance spectra were almost identical confirming that the key role in ensuring material conductivity is played by carbon nanotubes.

It is seen that the geopolymer material itself possesses a measurable conductivity even in a fully dry state. This is provided by the residues of moisture in the air-locked pores inside the material. The addition of the nanotubes causes the decrease of impedance by about 30% based on the comparison of the impedance module (Figure 3a). An increase of nanotube concentration up to 1% by mass reduces material impedance by about three times. The Nyquist spectra (Figure 3b) have the same form meaning that the general conductivity mechanism up to 1% nanotube contents remains the same. The phase shift of the AC signal in 1%-CNT samples differs from that of the 0.5% and CNT-free samples (Figure 3a). Its decrease means that the impact of the capacitive processes related to the electrolyte-filled pores becomes smaller with the increase of nanotube concentration. In other words, carbon nanotubes at high concentrations start playing the main role in charge delivery.

## 4. Discussion

The system of nanotubes randomly distributed in the geopolymer structure could not be presented by a simple equivalent electrical circuit composed of several physical elements. Therefore, our goal was to find a simple, averaged model to be used for the comparison of different samples and their conductive or capacitive properties.

The found solution included the application of an equivalent electrical circuit composed of frequency-dependent parameters. The fact that the parameter values are dependent on frequency may be confusing and non-consistent with the conventional concept of equivalent circuits. To explain this feature, we state that the parameters are estimated experimentally in a very narrow range of frequencies. A similar technique for fragmentary analysis was already proposed by Vladikova et al. [33,34,35] with respect to non-uniform research objects such as mesoporous oxide films.

Although the resistance and capacitance of individual physical elements do not depend on frequency, in the case of a complex system, such dependence may be valid and have a physical sense. The branches of an equivalent electrical circuit describe different paths of charge transport. The model resistors (Figure 4) describe the nanotubes. Although carbon is conductive, the nanotubes are very thin and are randomly distributed inside the material which results in total measured impedance of hundreds kOhms. The capacitors in a model describe the electrolyte areas between nanotubes that contain mobile metal ions (Ca^2+^, Na^+^, and others). They are enclosed in small pores. That disables them to move along the material but still lets them make a small oscillating movement to redistribute charge density, caused by applied alternating current. Such oscillation of local charge density provides internal capacitive properties of the material.

There are billions of charge paths in the material, yet the averaged equivalent circuit composed of three elements, *R*_s_ (series resistance), *R*_p_ (parallel resistance), and *C* (capacitance), stands for the average electrical response.

The nanotubes possess the electronic type of conductivity, while the conductivity of the space between them is provided by ions [36,37]. It is important to note that the whole material is not a conductor, i.e., it is not able to conduct direct electrical current due to the impossibility of charge transfer through the nanotube/geopolymer interface. The only electrical response can be detected at the application of an alternating voltage signal or a pulsed voltage signal, which causes a slight oscillation of ion concentration between the nanotubes.

The frequency of the alternating current signal is related to the time required for an ion to deliver a charge to the next “electrode”, the role of which is played by a carbon nanotube. Thus, the high-frequency signal, which gives a very small time, could proceed only through the small gaps between the conducting elements (Figure 4). A decrease in the signal frequency opens more paths for charge transport. The largest gaps would be enabled only at a considerably low frequency of the probing signal (Figure 4). The values of the physical parameters of the equivalent electrical circuit were calculated for the frequency range from 0.1 Hz to 30 Hz. The higher frequency points were not reliable for precise mathematical analysis due to high experimental data noise (Figure 3).

Within the fragmentary analysis method, the fragments of the spectrum are analyzed individually. i.e., first five points (from first to fifth) of the spectrum are taken and the values of *R*_s_, *R*_p_, and *C* are calculated. Then, the next five-point fragment, from the second through the sixth point, is considered. The results of the analysis of the spectra from Figure 3 are shown in Figure 5.

The selected number of points, five, was found to be optimal. From the fitting quality point of view, it is about two times more than the number of the calculated parameters (*R*_s_, *R*_p_, and *C*). On the other hand, it covers a frequency range, narrow enough to refer the data to one frequency in the middle of the fragment.

The CNT-free geopolymer has the lowest capacitance compared to the CNT-containing material (Figure 5a). The scatter of the points does not have any fundamental reason, it is caused by the noise of experimental data (Figure 3). Nevertheless, it still allows observing the common tendency of all the parameters to decrease with the frequency of the applied electrical signal. Both resistances are the smallest in the case of high CNT concentration and the largest for the CNT-free material, as could be expected taking into account the high electronic conductivity of the nanotubes. The gradual decrease of the resistance with frequency is distinctly notable in the case of *R_P_* (Figure 5c). The effect has the same reason as the fact that the geopolymer material is only penetrable by the alternating current. The mobile ions are only able to oscillate, but not to overcome long distances in one direction. Therefore, the high-frequency signal stimulates a larger amount of charge carriers to oscillate.

## 5. Conclusions

The presented research has shown the possibility of creating an innovative, ecological, relatively cheap, functional building material based on potentially hazardous industrial waste. A geopolymer based on metakaolin and CRT glass was reinforced with ultra-long multi-walled carbon nanotubes, which allowed to achieve growth of electrical conductivity over three times with only 1% nanotube mass content. Furthermore, a complex analysis of impedance spectra was accomplished to reveal not only conductive but also the intrinsic electrical capacitive behavior of the studied material. The addition of carbon nanotubes did not visibly affect the mechanical properties of examined material. Both flexural and compressive strength were high enough to consider the geopolymer reinforced with UL-MWCNTs for structural purposes. The presented technique can be regarded as a potential tool for monitoring the building constructions, with a promising application to the carbon nanotube modified geopolymer material.

## Figures and Tables

**Figure 1 materials-15-03543-f001:**
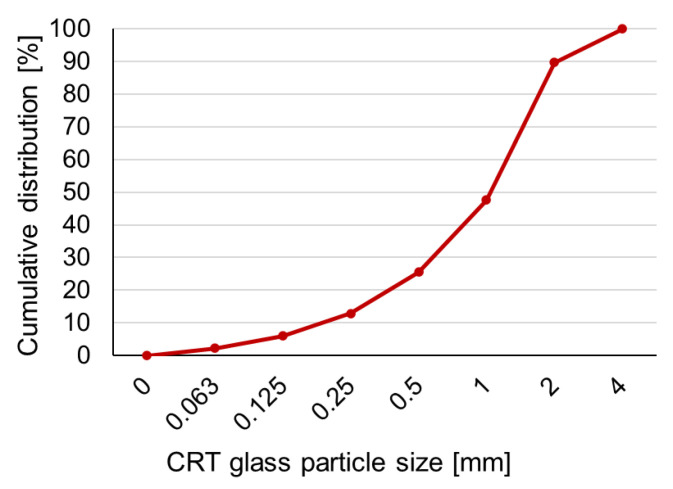
CRT glass particles size distribution.

**Figure 2 materials-15-03543-f002:**
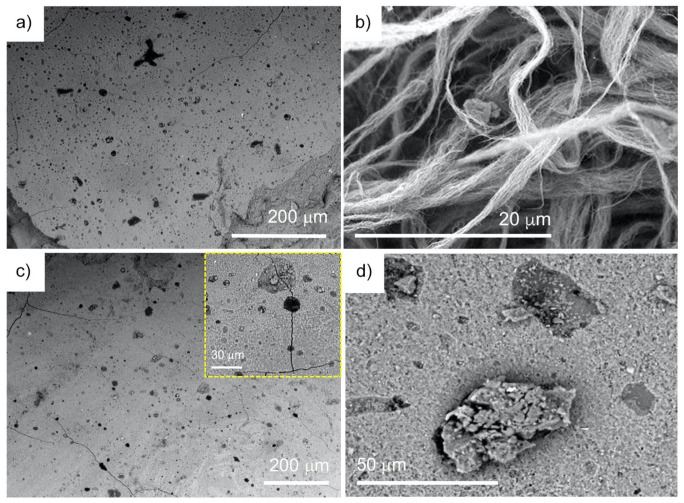
SEM images of: geopolymer (**a**), in-house MWCNTs (**b**), geopolymer nanocomposites containing 0.5 wt.% and 1 wt.% of in-house MWCNTs (**c**,**d**); the insets correspond to the indicated magnifications revealing the nanotube agglomerates.

**Figure 3 materials-15-03543-f003:**
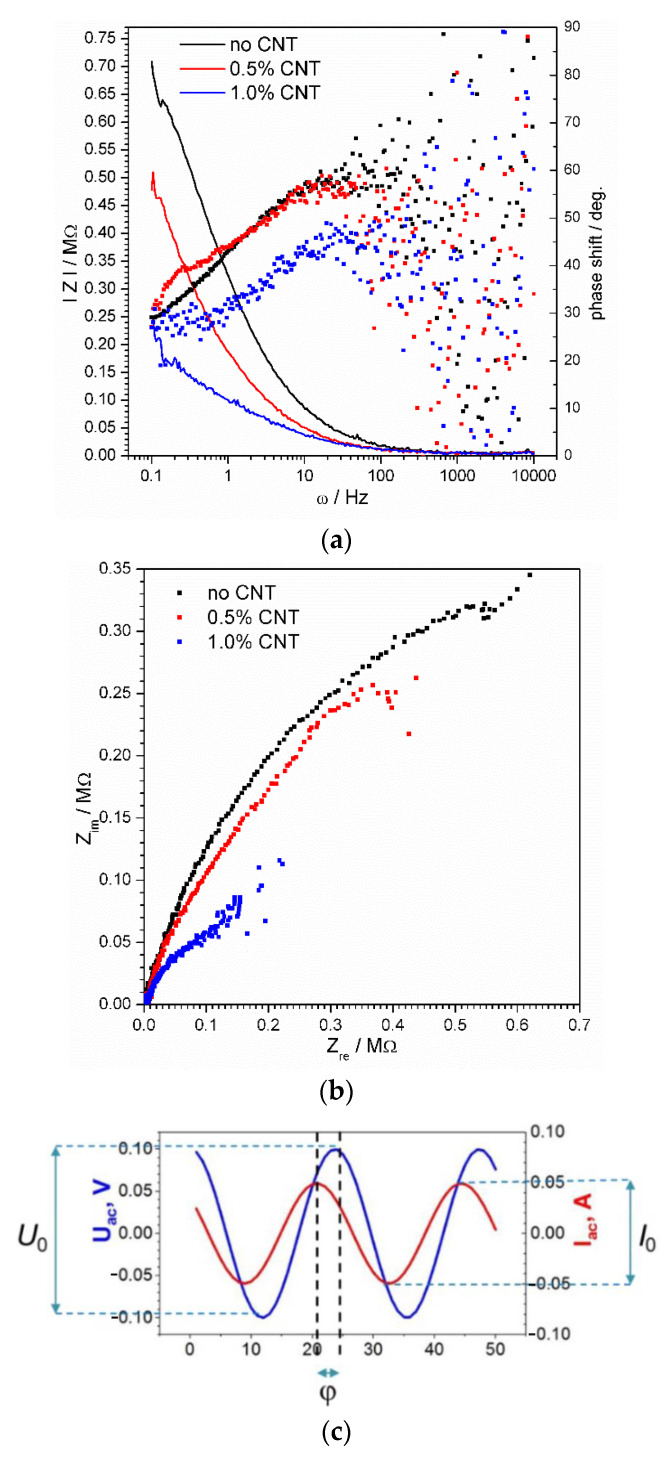
Bode (**a**) and Nyquist (**b**) spectra of geopolymer material modified with carbon nanotubes. Impedance module plot is shown by lines, while phase shift—by dots (**a**). A schematic illustration of the phase shift and amplitudes of the applied harmonic current signal and registered voltage response as functions of time (**c**).

**Figure 4 materials-15-03543-f004:**
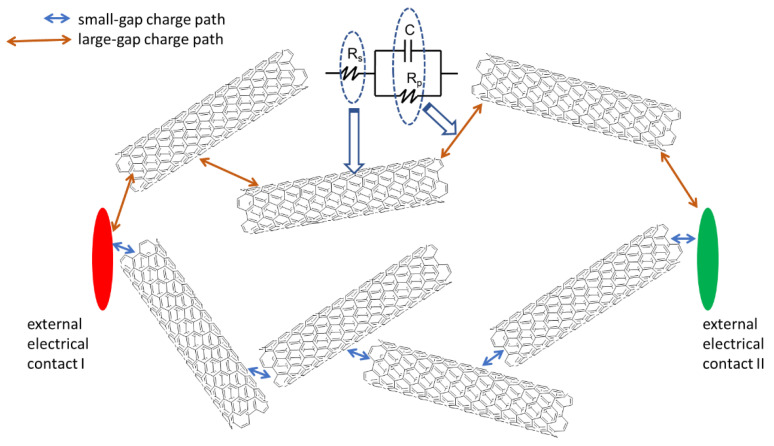
Illustration of two paths of charge transport that give a different electrical response. The two-side arrows symbolize the oscillation of charge carriers in the medium under external AC voltage exposure.

**Figure 5 materials-15-03543-f005:**
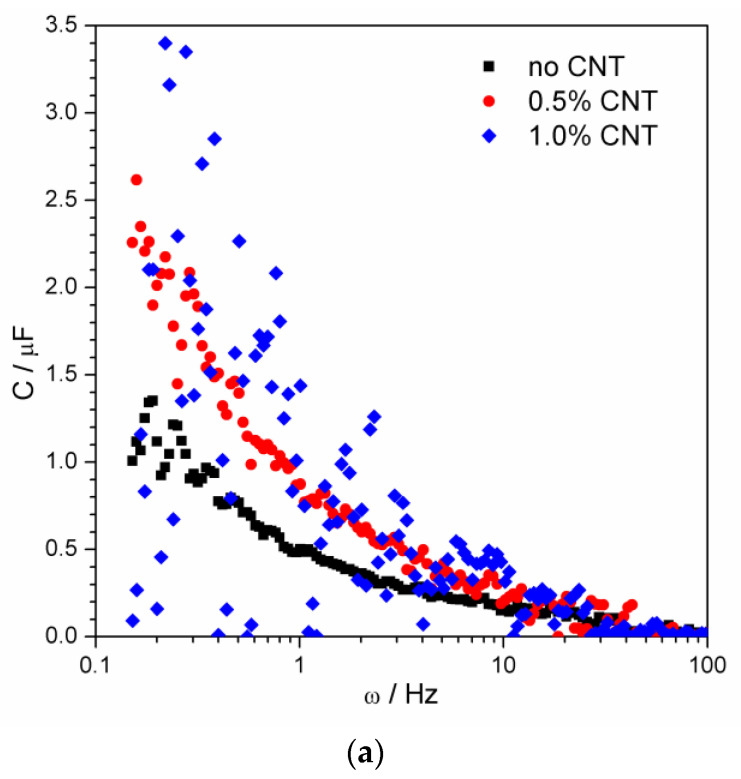
Calculated values of average capacitance *C* (**a**), series resistance *R_S_* (**b**), and parallel resistance *R_P_* (**c**) as a function of probing signal frequency.

**Table 1 materials-15-03543-t001:** Chemical composition (wt.%) of metakaolin ^1^ and CRT glass ^2^.

Oxide Composition	Metakaolin	CRT Glass
SiO_2_	53.12	76.10
Al_2_O_3_	42.14	1.37
K_2_O	0.73	2.36
TiO_2_	0.64	0.12
Fe_2_O_3_	0.45	0.38
CaO	0.44	5.24
MgO	0.26	1.64
H_2_O^−^	0.22	-
Na_2_O	0.09	6.25
P_2_O_5_	0.03	-
MnO	0.01	-
BaO	-	2.62
PbO	-	1.61
SrO	-	1.42
SO_3_	-	0.55
ZrO_2_	-	0.28
ZnO	-	0.05
As_2_O_3_	-	0.01

^1^ Data obtained from the producer: Astra Technologia Betonu^®^. ^2^ Determined by XRF analysis by EkotechLAB^®^.

**Table 2 materials-15-03543-t002:** Mixtures composition.

Mixture	Metakaolin (kg/m^3^)	CRT Glass(kg/m^3^)	Sodium Silicate (kg/m^3^)	Sodium Hydroxide (kg/m^3^)	Carbon Nanotubes (kg/m^3^)
No CNT	898	898	449	225	-
0.5% CNT	898	898	449	225	4.49
1.0% CNT	898	898	449	225	8.98

**Table 3 materials-15-03543-t003:** Compressive and flexural strength of geopolymer.

	no CNT	0.5% CNT	1.0% CNT
Flexural strength (MPa)	6.2	5.3	5.8
Compressive strength (MPa)	66.6	60.7	59.0

**Table 4 materials-15-03543-t004:** Density of samples.

Density	No CNT	0.5% CNT	1.0% CNT
(kg/m^3^)	2010	1970	1970

## Data Availability

Data available upon request.

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
