# Peer review of "Electrical Properties of the Carbon Nanotube-Reinforced Geopolymer Studied by Impedance Spectroscopy"

_materials, 2022, doi:10.3390/ma15103543_

Round 1

Reviewer 1 Report

The paper focused on the Electrical properties of the carbon nanotube-reinforced geopolymer studied by impedance spectroscopy. The authors explored the effects of different amount of carbon nanotube addition on the conducitive of the geopolymer. The content is interesting. But there are also some obvious shortcomings. So my suggestion is “accepted after a major revision”.

  1. About the title of this paper, “Electrical and mechanical properties of the carbon nanotube-reinforced geopolymer studied by impedance spectroscopy” is not accurate. In fact, this paper focuses on the electrical properties of the carbon nanotube-reinforced geopolymer with a little decreased mechanical properties studied by impedance spectroscopy.
  2. About abstract, “Due to its good conductive properties”, this expression is not proper. The conductive properties of the traditional geopolymer is not good, as is also revealed in the context of this paper, such as “The registered conductivity of the modified geopolymer material, though it is very small, is a unique feature” in Page 7. Just for this reason, many researchers added carbon nanotube or carbon fiber, etc.
  3. About the discussion, the authors discuss two types of conductivity, which are expressed by “The nanotubes possess the electronic type of conductivity, while the conductivity of the space between them is provided by ions”. Here should be added some references.

Author Response

Responses to the Reviewers

Reviewer 1

The paper focused on the Electrical properties of the carbon nanotube-reinforced geopolymer studied by impedance spectroscopy. The authors explored the effects of different amount of carbon nanotube addition on the conductive of the geopolymer. The content is interesting. But there are also some obvious shortcomings. So my suggestion is “accepted after a major revision”.

Thank you for your kind reception of our paper and your very helpful comments. We have tried to change the text accordingly. Please find below our answers to your comments.

  1. About the title of this paper, “Electrical and mechanical properties of the carbon nanotube-reinforced geopolymer studied by impedance spectroscopy” is not accurate. In fact, this paper focuses on the electrical properties of the carbon nanotube-reinforced geopolymer with a little decreased mechanical properties studied by impedance spectroscopy.

Thank You for this comment. The idea of the initial title was to give a general scope of the article. Its current version suggests that not only electrical, but also most important mechanical properties have been checked, although it does not imply any extraordinary mechanical properties, so we decided to follow your suggestion and remove “mechanical” from the title.

  1. About abstract, “Due to its good conductive properties”, this expression is not proper. The conductive properties of the traditional geopolymer is not good, as is also revealed in the context of this paper, such as “The registered conductivity of the modified geopolymer material, though it is very small, is a unique feature” in Page 7. Just for this reason, many researchers added carbon nanotube or carbon fiber, etc.

Thank You for drawing attention to that sentence. It has been changed by “Due to the registrable electrical conductivity, this material also attracts the interest of other fields of science and industry as a potential functional material.”

  1. About the discussion, the authors discuss two types of conductivity, which are expressed by “The nanotubes possess the electronic type of conductivity, while the conductivity of the space between them is provided by ions”. Here should be added some references.

The type of conductivity is defined during general knowledge of solid-state physics and the chemical structure of the materials. Nanotubes do not incorporate any mobile species, thus their only conductivity type could be electronic. The geopolymer material is composed of oxides and salts, and has no components that could behave as metals in charge conduction, so ions are the only charged mobile species that are responsible for conductivity.

Following your suggestions we have added references:

  1. Ebbesen, T.W.; Lezec, H.J.; Hiura, H.; Bennett, J.W.; Ghaemi, H.F.; Thio, T. Electrical Conductivity of Individual Carbon Nanotubes. Nature 1996, 54–56, doi:10.1038/382054a0.
  2. Du, F.; Scogna, R.C.; Zhou, W.; Brand, S.; Fischer, J.E.; Winey, K.I. Nanotube Networks in Polymer Nanocomposites: Rheology and Electrical Conductivity. Macromolecules 2004, 37, 9048–9055, doi:10.1021/ma049164g.

Reviewer 2 Report

The paper studies the electrical and mechanical properties of the carbon nanotube reinforced geopolymer which can be used as a smart material for Structural Health Monitoring. This manuscript presents some experimental work but lacks in-depth analysis. Detailed comments are shown below for your information.

  1. The number of samples in this paper is small, resulting in low  reliabilityof  The sample containing CNT has only one bending test data and two compressive strength data at each dosage. Therefore, it is impossible to determine whether the test results in Figure 2 are test rules or experimental errors. 
  2. The carbon nanotubes in 3are bundles. How is the dispersion of 0.5% and 1% carbon nanotubes in the sample? Electron microscopic images also indicate the agglomeration of CNTS, but it should be indicated where the agglomeration occurs.
  3. Humidity and temperature have great influence on the AC impedance spectrum data, and it is mentioned in this paper that humidity has almost no influence on it, so the corresponding original data should be given.
  4. Rs, Rp and C values at different frequencies are givenin this paper, but AC impedance spectroscopy is generally aimed at transient analysis, which means that different electrical polarization processes occur at different frequencies. What is the significance of Rs, Rp and C changes with frequency?
  5. Lines 260 to 262, the authors say that although carbon conducts electricity, the nanotubes are very thin and randomly distributed, resulting in a total resistance of several hundred ohms.Please give the basis on which the resistance is this order of magnitude.
  6. The author gives the circuit diagram, but all the circuit diagram involves is the specimen itself. How to consider the interface between the specimen and the electrode?Is it the same as the geopolymer-carbon nanotube interface?
  7. In general resistance is independent of frequency.On what basis does the author suggest that resistance varies with frequency (FIG. 6)?
  8. The author obtained Figure 6 by fitting every 5 points in Figure 4.What is the basis of 5 points fitting? With such a small number of points, are the parameters obtained reliable?

Author Response

Reviewer 2

The paper studies the electrical and mechanical properties of the carbon nanotube reinforced geopolymer which can be used as a smart material for Structural Health Monitoring. This manuscript presents some experimental work but lacks in-depth analysis. Detailed comments are shown below for your information.

Thank you for your kind reception of our paper and your very helpful comments. We have tried to change the text accordingly. Please find below our answers to your comments.

  1. The number of samples in this paper is small, resulting in low reliability of the sample containing CNT has only one bending test data and two compressive strength data at each dosage. Therefore, it is impossible to determine whether the test results in Figure 2 are test rules or experimental errors.

Thank you for your remark. We do realize that the number of samples was too small for the proper mechanical behaviour determination. Our goal was to preliminary assess if CNTs are influencing mechanical behaviour. We are going to extend mechanical tests in the future. However, we did not emphasise it in the paper. We have added the appropriate explanations in the section describing strength results and modified that section also.

  1. The carbon nanotubes in 3 are bundles. How is the dispersion of 0.5% and 1% carbon nanotubes in the sample? Electron microscopic images also indicate the agglomeration of CNTS, but it should be indicated where the agglomeration occurs.

Thank you for your remark. Nanomaterial was applied in dry form, it was not in dispersion. We have tested this in other, not described in this paper tests, and for geopolymer, there were no significant changes for both types of nanomaterial addition, so seeking the easiest applicable form, we decided to use dry CNTs.

In the absence of creating a three-dimensional network by means of CNTs (see micrographs), we should not expect excellent conductivity. However, it does not diminish the potential of applications of even such imperfect systems as the so-called 'smart composites' - that's the question of selecting the measurement/amplification system.

  1. Humidity and temperature have a great influence on the AC impedance spectrum data, and it is mentioned in this paper that humidity has almost no influence on it, so the corresponding original data should be given.

Thank you for the important remark. We do realize the effects of moisture and temperature influences on the readings, we have made numerous research connected with these phenomena, described in our former papers. In this paper, as we mentioned in the text, we have compared the impedance of the same samples after dipping in the water and after drying. The wet sample was gently dried with a towel to remove water from the surface and avoid electrode contact with water. No noticeable change was noticed. The Figure (in attached file) shows the equality of Nyquist spectra for the wet and dry samples. We have decided that presenting this graph is not useful and just left the description in the text.

  1. Rs, Rp and C values at different frequencies are given in this paper, but AC impedance spectroscopy is generally aimed at transient analysis, which means that different electrical polarization processes occur at different frequencies. What is the significance of Rs, Rp and C changes with frequency?

Thank you for your question.

Actually, impedance spectroscopy is much more useful for the study of a sample in a steady-state rather than during its change. The transient analysis is not possible in the case of the basic impedance spectroscopy method. There are advanced modifications of this method, called potentiodynamic or dynamic impedance spectroscopy (PDEIS), that allow for ultrafast registration of the spectra of an unstable object.

The parameters change with frequency does not arise from the object evolution. It was the only way, proposed by us, to consider the electrical behaviour within the scope of resistance and capacitance elements. One may not expect that such a complicated object would be described by an equivalent electrical circuit consisting of several parameters, as it happens in the case of uniform, homogeneous systems. Thus, the proposed analysis method is the only possible solution to describe a complicated solid-state object with randomly distributed conductive dopants.

A further explanation of the frequency dependence of parameters is given with the answer to your 7th comment.

  1. Lines 260 to 262, the authors say that although carbon conducts electricity, the nanotubes are very thin and randomly distributed, resulting in a total resistance of several hundred ohms. Please give the basis on which the resistance is this order of magnitude.

Thank your for your remark. It was said in the manuscript, that the resistance of carbon nanotubes must be about the order of 100 kΩ. This value is based solely on experimental results with no reference to the electrical properties of the nanotubes. According to Figure 4, the real and imaginary components of the impedance range from 1kΩ up to 250kΩ (for 1% CNT mass content). The cited piece of discussion only gives a possible explanation for a high impedance despite the incorporation of conducting dopants.

Nevertheless, we admit, that the word “resistance” should be replaced with “impedance” in the mentioned text fragment.

  1. The author gives the circuit diagram, but all the circuit diagram involves is the specimen itself. How to consider the interface between the specimen and the electrode? Is it the same as the geopolymer-carbon nanotube interface?

We made several tests to reveal any effect of the electrode-sample interface on the total system response. The electrode area, material (steel, aluminium and copper) and pressing force were varied with no effect on the resulting spectra. That led us to a conclusion that the impedance of the electrode-sample interface is negligibly small compared to the impedance of the solid material. Being a series element in the charge path, the interface impedance remains non-extractable from the response.

  1. In general resistance is independent of frequency. On what basis does the author suggest that resistance varies with frequency (FIG. 6)?

Thank you for your question.

In classical impedance spectroscopy analysis, all the elements, including resistances and capacitances, are independent of frequency. However, if following that rule, the geopolymer-CNT macroscopic object would require an enormous amount of frequency-independent elements to describe all the possible charge paths inside the material. The idea of introducing a frequency-dependent parameter has been developed to the average response of similar charge paths and give a quantitative characteristic of the material.

As an addition to this explanation, please see the answer to remark 4 and Figure 5 and the corresponding comments in the text.

  1. The author obtained Figure 6 by fitting every 5 points in Figure 4. What is the basis of 5 points fitting? With such a small number of points, are the parameters obtained reliable?

Thank you for your question.

The selected number of points, five, was found to be optimal. From the fitting quality point of view, it is about two times more than the number of varied parameters (equal 3, i.e. Rs, Rp and C). On the other hand, it covers a frequency range, narrow enough to refer the data to one particular frequency. With the chosen number of experimental points, i.e. 20 points per decade on a logarithmic scale, the five-point interval covered a range between w/1.26 and w*1.26, where w is the middle frequency of the considered range; the parameters calculated for a narrow interval were attributed to the frequency w.

Reviewer 3 Report

The topic of the paper is interesting and the paper is very well structured. The novelty of the paper is well stated and the question that authors tried to address in their research has been fully addressed. However, the following are required to be considered by the authors before the paper being accepted.   

  • The current conclusions are massive, and it contains lots of unnecessary information. Therefore, authors are firstly required to state the main aim of the current research at the conclusion before listing the main findings. Then the findings should be summarised in and shortened as they are seems to be too much information there. 

Author Response

RESPONSES TO REVIEWERS

Reviewer 3

The topic of the paper is interesting and the paper is very well structured. The novelty of the paper is well stated and the question that authors tried to address in their research has been fully addressed. However, the following are required to be considered by the authors before the paper being accepted.  

The current conclusions are massive, and it contains lots of unnecessary information. Therefore, authors are firstly required to state the main aim of the current research at the conclusion before listing the main findings. Then the findings should be summarised in and shortened as they are seems to be too much information there.

Thank you for your kind reception of our work and paper. We are truly grateful.

We have structured the conclusions in chapter 5. of our paper, in order to make them useful and provide a brief summary of described research and obtained results. Shall you have any detailed recommendation regarding this part, we will be happy to correct this part accordingly. 

Round 2

Reviewer 2 Report

No further comments